# Muscle functions as a connective tissue and source of extracellular matrix in planarians

Lauren E. Cote[1], Eric Simental [1,2] & Peter W. Reddien[1]

Regeneration and tissue turnover require new cell production and positional information. Planarians are flatworms capable of regenerating all body parts using a population of stem cells called neoblasts. The positional information required for tissue patterning is primarily harbored by muscle cells, which also control body contraction. Here we produce an in silico planarian matrisome and use recent whole-animal single-cell-transcriptome data to determine that muscle is a major source of extracellular matrix (ECM). No other ECM-secreting, fibroblast-like cell type was detected. Instead, muscle cells express core ECM components, including all 19 collagen-encoding genes. Inhibition of muscle-expressed *hemicentin-1* (*hmcn-1*), which encodes a highly conserved ECM glycoprotein, results in ectopic peripheral localization of cells, including neoblasts, outside of the muscle layer. ECM secretion and *hmcn-1*-dependent maintenance of tissue separation indicate that muscle functions as a planarian connective tissue, raising the possibility of broad roles for connective tissue in adult positional information.

[1] Howard Hughes Medical Institute, Whitehead Institute, and Department of Biology, Massachusetts Institute of Technology, 455 Main St, Cambridge, MA 02142, USA. [2] Present address: University of California San Francisco, 600 16th Street, San Francisco, CA 94143, USA. Correspondence and requests for materials should be addressed to P.W.R. (email: reddien@wi.mit.edu)

The ability to regenerate missing body parts requires both sources of new cells and adult patterning processes to specify and organize new tissues. How positional information is maintained and transmitted during adult regeneration remains poorly understood. Planarians are flatworms capable of regenerating any missing body part and are a model system well suited for studying patterning and maintenance of tissue architecture during regeneration and tissue turnover[1,2].

Planarians possess a population of adult dividing somatic cells called neoblasts that mediate constant tissue turnover associated with longevity and regeneration. Neoblasts divide to give rise to all new cells in the animal[3]. The neoblast cell population includes pluripotent stem cells[4,5] and numerous classes of specialized neoblasts that are progenitors for specific differentiated tissues[1]. In vertebrates, lineage-restricted progenitors and/or dedifferentiation can produce new cells during regeneration[6]. Planarian neoblasts exclusively reside in the parenchyma, a dense tissue compartment with sparse ECM surrounding internal organs such as the central nervous system and the intestine[3].

Patterning information from differentiated tissues regulates planarian regeneration and tissue turnover, including neoblast specification and the targeting of neoblast-derived progenitor cells to the correct locations within the animal[7–10]. Multiple secreted and transmembrane molecules have been identified that are constitutively expressed and are required for the regeneration and maintenance of region-appropriate tissues in planarians[1]. Inhibition of these genes with RNA interference (RNAi) causes the regeneration of regionally incorrect tissues or duplication of regional anatomy. For example, Bmp pathway inhibition leads to ventralization whereas Wnt pathway inhibition results in the formation of a head instead of a tail at posterior-facing amputation planes[1,2]. Genes that display constitutive regional expression and that are associated either with a patterning phenotype or with patterning pathways are called position control genes (PCGs)[7]. PCG transcriptional expression domains include anterior, posterior, and medial-lateral gradients[11,12] and these expression domains primarily reside in one major cell type: muscle[7,13]. Most PCGs are strongly expressed in the body wall muscle, which encases the animal. Expression of a subset of PCGs also occurs in dorsal-ventral muscle and intestinal muscle[7,13–15]. This observation led to the hypothesis that planarian muscle harbors a positional information coordinate system guiding tissue turnover and regeneration[7].

We sought to explore the attributes of muscle that might make it well suited to act as a source of positional information by analyzing the planarian muscle transcriptome. In addition to actomyosin contractility genes, which are required for body integrity and movement[13,16], a canonical marker of planarian muscle encodes a fibrillar collagen chain[7]. Collagens are some of the most abundant proteins in animals, comprising approximately one-third of all protein by mass in humans[17]. These triple-helical trimeric proteins assemble to form fibrous bundles that are the major structural components of the extracellular matrix (ECM). The ECM is the scaffold formed from insoluble fibrils and gel-like material that surrounds cells and tissues. Among its many important developmental roles, ECM functions to sequester and present signaling molecules and to separate tissues from one another[18–20]. Most planarian ECM that is clearly visualized by electron microscopy (EM) is found within the sub-epidermal membrane as dense filaments that are ~10 nm in diameter[21]. Prior EM studies suggested that planarian muscle cells might secrete collagenous filaments and glycosaminoglycan-positive material because of the close proximity of the sparse ECM in the parenchyma to the dorsal-ventral muscles and the presence of secretory vesicles found in body wall muscle cells near the sub-epidermal membrane[21,22].

The secreted proteins that comprise the ECM of an animal are collectively called the matrisome[23,24]. Vertebrate matrisome proteins are partitioned into (1) core matrisome components such as collagens and large glycoproteins that are heavily decorated with carbohydrates including glycosaminoglycans and (2) ECM-associated proteins, which are secreted proteins that interact with and regulate the ECM but that are not major structural ECM components. In vertebrates, one principal source of ECM components is fibroblasts, which are spindle-shaped cells found in the connective tissue that supports and surrounds various tissues[25,26]. Other cellular sources of ECM include endothelial cells, immune cells, smooth muscle, and epidermis. Major invertebrate model organisms such as Drosophila and C. elegans secrete major ECM components from haemocytes and body wall muscle, respectively[27]. However, the identity of cells broadly responsible for ECM secretion has been poorly studied across major clades of the metazoans, such as the Spiralia, hindering broader understanding of the evolution of connective tissue.

Connective tissues broadly function to support other tissues, by binding, separating, and connecting them, often through ECM formation. We reasoned that whichever cells predominantly express ECM proteins should comprise the connective tissue of planarians. In this study, we use organism-wide single-cell transcriptome analyses and determine that planarian muscle is the major source of core ECM components, suggesting that it functions as a connective tissue for planarians. Supporting this hypothesis, a gene encoding a highly conserved glycoprotein, Smed-hemicentin-1 (hmcn-1), is expressed specifically in body wall muscle cells and is required to maintain the physical separation of the inner planarian parenchymal tissue from the outer epidermis. The combined positional information and connective tissue role of planarian muscle is significant for understanding the evolutionary origins of animal connective tissues and the roles of connective tissue in promoting proper tissue architecture in animal regeneration and tissue turnover.

## Results

**in silico identification of the planarian matrisome.** To identify the cells that are major sources of planarian ECM and to explore the hypothesis that muscle is such a source, we first sought to systematically identify components of the planarian matrisome. Prior work characterizing human, mouse, and zebrafish matrisomes has established a computational approach for identifying matrisome components in different species[23,24,28,29]. By utilizing domain annotations from prior knowledge and mass spectrometry data of solubilized mouse and human ECM, Naba and colleagues identified a set of InterPro domains that strongly predict ECM localization[24]. In addition, this work categorized matrisome proteins into core matrisome components and ECM-associated proteins. The core matrisome includes all major structural components of the extracellular matrix and is a highly curated list containing collagens, numerous glycoproteins (e.g., nidogen, laminin, and fibronectin), and proteoglycans, which are glycoproteins post-translationally modified with glycosaminoglycans such as heparan sulfate (e.g., perlecan (plc)). The ECM-associated proteins include ECM-affiliated proteins (e.g., glypicans, GPI-anchored proteins), ECM regulators (e.g., matrix metalloproteases (MMPs), peptidases), and secreted factors (e.g., Wnt, TGF-beta signaling ligands). Planarians are surrounded by a soft cellular epidermis and correspondingly do not have homologs of specialized ECM components associated with an exoskeleton such as nematode cuticlins or insect chitins. We utilized human matrisome data to identify predicted planarian matrisome components.

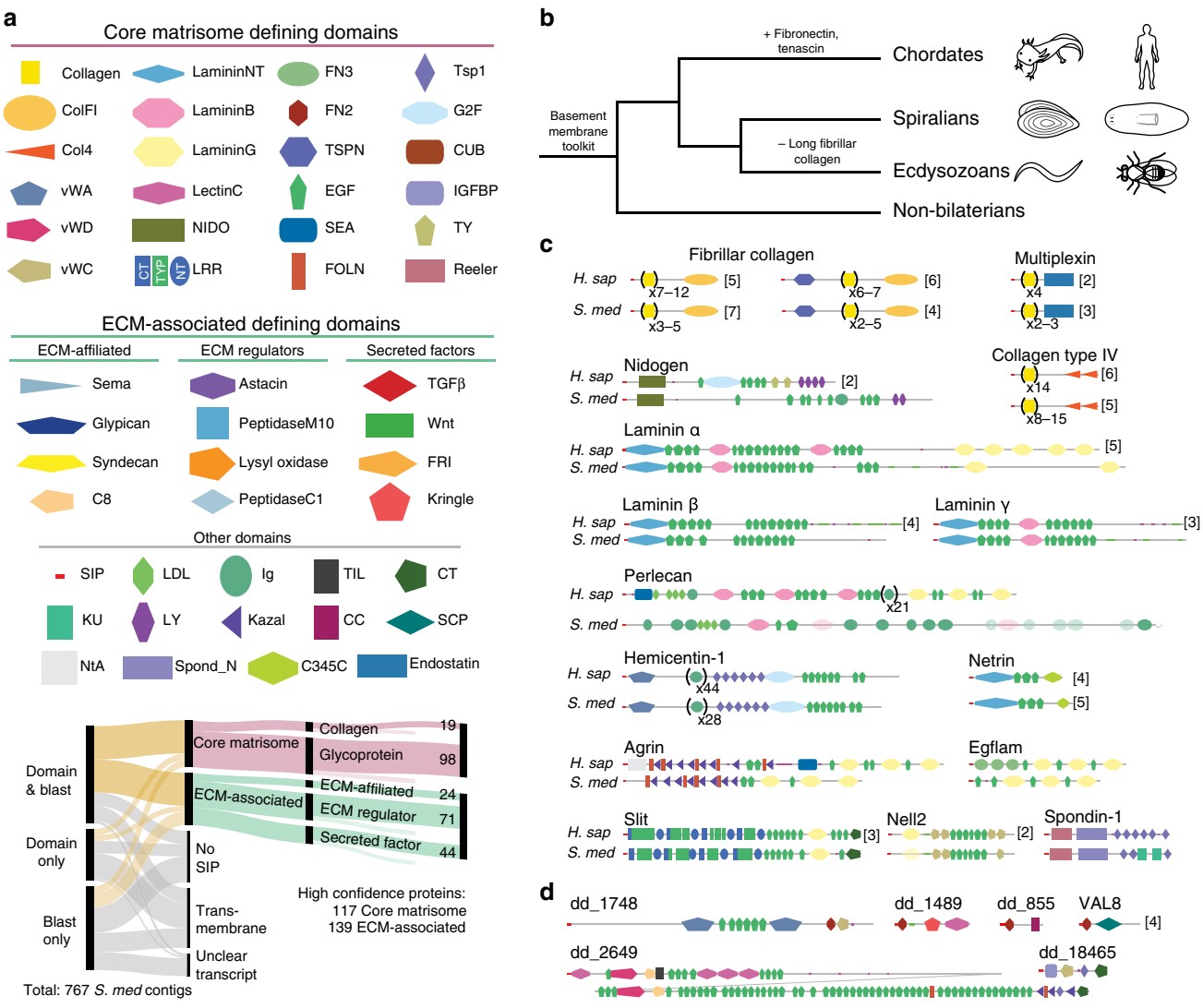

**Fig. 1** The planarian matrisome includes proteins with highly conserved domain structures. **a** Domains present in both planarians and humans that define the matrisome[24] were used, along with blastx hits to human matrisome proteins, to categorize the ~750 contigs as shown and define the planarian matrisome. Light colored lines indicate low confidence in ECM localization. SIP, signal peptide. **b** Phylogenetic relationship between planarians and other model organisms showing the ancient origin of basement membrane proteins and gain or loss of important ECM proteins. **c** Domain architectures, colored as in Fig. 1a, of core matrisome proteins which are conserved between planarians and humans. Domains in parentheses have indicated number of repeats. Numbers in square brackets indicate number of proteins in *Schmidtea mediterranea* (*S. med*) or *Homo sapiens* (*H. sap*) with indicated domain architecture. **d** Domain architectures found in the planarian core matrisome not found in humans

To identify planarian genes encoding predicted proteins from each category of the matrisome (matrisome genes), we searched for predicted *Schmidtea mediterranea* transcripts that were annotated[30,31] with matrisome-defining InterPro domains and did not contain an excluding domain such as a kinase domain (Eval <0.1, 491 contigs, Methods). Sixty-four out of 93 matrisome-defining InterPro domains found in humans were present in proteins encoded by the planarian transcriptome (Fig. 1a). We used tblastn and blastx to identify planarian proteins encoded by the planarian transcriptome that were similar to complete or partial human matrisome proteins (Eval <0.01, 597 contigs). We then applied a set of filters to pare down this set of 767 total contigs to those genes encoding proteins predicted to be secreted and to be localized to the ECM (Fig. 1a, Supplementary Data 1, 2, Methods). First, we used gene predictions from genomic sequence and manual inspection of RNA-sequencing read density[31] to find the longest coding sequence of genes. Then, we checked transcripts for the presence

of sequence encoding a signal peptide. Finally, to categorize each planarian CDS into those encoding predicted core matrisome or ECM-affiliated proteins, we examined the human best blastx annotation for each gene and the predicted domain structure of the encoded protein. We supplemented the list of identified secreted factors with genes encoding homologs of Noggin/ Noggin-like proteins and Notum[1]. This in silico approach resulted in the identification of 133 planarian genes encoding predicted core matrisome proteins and 167 genes encoding predicted matrisome-associated proteins (Supplementary Data 1, Supplementary Fig. 1).

**Domain organizations of the planarian core matrisome proteins**. Our computational approach to annotating the planarian matrisome found evidence for the presence of components of the basement membrane toolkit found in all metazoans[32] and no evidence for homologs of ECM proteins

considered vertebrate-specific, such as fibronectin (Fig. 1b). Many putative matrisome proteins encoded by the planarian genome did not have a clear human homolog; therefore, an ECM-localization confidence rating was assigned to each protein. High-confidence matrisome components were those with a signal peptide and one or more predicted matrisome-defining domains (Fig. 1c, d, Supplementary Data 1). Within the set of 256 high confidence planarian matrisome proteins, 104 had a domain organization that was highly similar to a human matrisome protein, including 23 core matrix glycoproteins and 19 collagens (Fig. 1c). The planarian components of the highly conserved basement membrane toolkit included a single clear homolog of nidogen, each laminin chain[14], two agrin homologs, and five clear collagen type IV proteins (Fig. 1c, Supplementary Fig. 1). The planarian homolog of the heparan sulfate proteoglycan (plc) has similar domain architecture to the N-terminal portion of human plc, but lacks C-terminal lamininG-EGF repeats, similar to multiple isoforms of *C. elegans* plc homolog UNC-52. Another gene encoding a putative *S. mediterranea* plc homolog (dd_8356), is a pharynx marker previously termed *laminin*[33]. Because this protein has a similar N-terminal domain structure to plc but ends after the first lamininB domain we have renamed this gene *pharynx muscle-1* (*phmus-1*).

In addition to proteins with a clear domain structure similar to a human matrisome protein, other high confidence planarian matrisome proteins had a domain organization similar to the extracellular domain of a receptor or had an organization of matrisome domains not found in humans (Fig. 1d). For example, five large genomic regions were identified that contain coding sequence for putative large ECM proteins with VWD-C8-TIL repeats followed by a large (~500–2000 amino acids) highly repetitive region, and ending in vWC, TSP, and CT domains (Fig. 1d, Supplementary Fig. 1). Although the N and C-termini of these predicted proteins show some similarity to the domain architecture of SCO-spondins, these are likely related to mucins, such as human Mucin-2, given their repetitive central region.

The high confidence planarian ECM-affiliated proteins included multiple mucins, NCAM, glypican, and GPI-anchored anosmin proteins (Supplementary Data 1). The second largest ECM category found was of 71 potential ECM regulators containing multiple homologs of Tolloid-like, ADAMTS, four previously characterized MMPs[34–37], and other peptidases. The ShKT domain is present in sea anemone toxins and is common in nematode peptidases[38], and was found within six predicted planarian ECM regulators. Finally, the planarian secreted factor category contained multiple well-studied planarian Wnt and TGF-beta family signaling ligands[1,2].

In contrast to the high-confidence matrisome proteins, low-confidence matrisome components have a matrisome-defining domain but some characteristics that might indicate intracellular localization. Several low-confidence matrisome components had matrisome-defining domains without a clear signal peptide (5/44 proteins) and nine genes encoded small (<200 amino acid) predicted secreted factors with weakly identified domains. The low confidence matrisome proteins included 15 cathepsin enzymes, which are canonically found within the lysosome, and some of which have been characterized as planarian gut proteases[39]. Six of the low confidence core matrisome genes encode proteins with an Ig-LDL domain structure that is Platyhelminth-specific and similar to the very N-terminal portion of plc (Supplementary Fig. 1). Although Ig and LDLa domains are often present in matrisome proteins, they are not considered matrisome-defining domains because of their presence in intracellular/transmembrane proteins such as the muscle scaffolds titin and obscurin, and cholesterol receptors. Predicting the localization of these proteins to the

ECM was unclear and they were therefore classified as low-confidence matrisome proteins.

In sum, we created an in silico definition of the planarian matrisome that identifies a large set of proteins with a complete CDS that can be utilized to investigate the nature and role of ECM in tissue homeostasis and regeneration. The curated core matrisome list of genes encoding high-confidence collagens and glycoproteins is of particular interest for understanding which cell types are a major source of planarian ECM and for defining their roles.

**Matrisome expression specificity in cell-type atlases.** Recently, extensive single cell sequencing (SCS) has been applied to characterize the cell-type transcriptomes of planarians[5,12,40,41]. Application of this approach systematically to each region of the planarian body reached saturation-level coverage of every cell type to generate a complete *S. mediterranea* cell-type atlas[12]. To determine which cell type(s) have major roles in secreting ECM, we utilized these data to search for the cell types that express genes encoding planarian matrisome proteins.

Similarities in gene expression of all sequenced cells from the SCS data can be visualized in clusters in a two-dimensional tSNE plot where each dot represents a single cell (Fig. 2a). To facilitate visualization, cells were assigned to one of nine broad planarian tissue-type classes (Fig. 2a), including both terminally differentiated cells and progenitors for those cells generated by neoblasts. For any given cell type, we can determine its complement of specifically expressed genes and for any specific gene, assess whether it has specific cell-type-enriched expression, as determined by the area under receiver operating characteristics curve (AUC) value closest to 1.0 (Fig. 2b). Gene set enrichment analysis (gsea) showed that muscle is enriched in the expression of actomyosin contractile machinery proteins (gsea normalized enrichment score (NES) 1.83; $p_{adj} = 0.00089$) and expresses both smooth and striated muscle markers, as expected[42]. Planarian muscle cells were enriched in the expression of core matrisome collagen-encoding genes (NES 2.05; $p_{adj} = 0.00089$) and multiple matrisome genes displayed expression specifically in muscle cells (Fig. 2b, Supplementary Fig. 2–4). This result is consistent with the observation that one fibrillar collagen-encoding gene, *colF-2*, was co-expressed in cells expressing muscle-specific actomyosin components *troponin* and *tropomyosin*[7].

In fact, all 19 planarian genes predicted to encode collagens were expressed in muscle cells with a high degree of specificity (Fig. 2b, Supplementary Data 1, Supplementary Figs. 2–5). The planarian genome contains 11 genes encoding complete fibrillar collagens, containing the COLFI C-terminal trimerization domain that aids in collagen fibrillogenesis[17]. Four of these proteins contain a TSPN/LamG N-terminal domain and one contains a VWC N-terminal domain (Supplementary Data 1). Nine of these fibrillar collagens were expressed specifically in all differentiated muscle cell types, *colF-11* was highly expressed in pharynx muscle, and *colF-10* had restricted expression to body wall muscle (Supplementary Fig. 4)[12]. Five appreciably expressed planarian *collagen* genes encode type IV collagens (*col4–1–5*), which are basement membrane-specific collagen proteins that form a meshwork instead of long fibrils. Two other putative genes encoding collagen type IV (*col4–6* and *col4–7*) exist within the planarian genome, but were minimally detectable in RNA-sequencing data and were not considered high confidence collagen-encoding genes (Supplementary Data 1). Unlike in many organisms, the planarian genes encoding collagen type IV are not arranged in a head-to-head orientation in the genome . By double fluorescence in situ hybridization (FISH), genes encoding collagen type IV chains were highly expressed in all muscle cells,

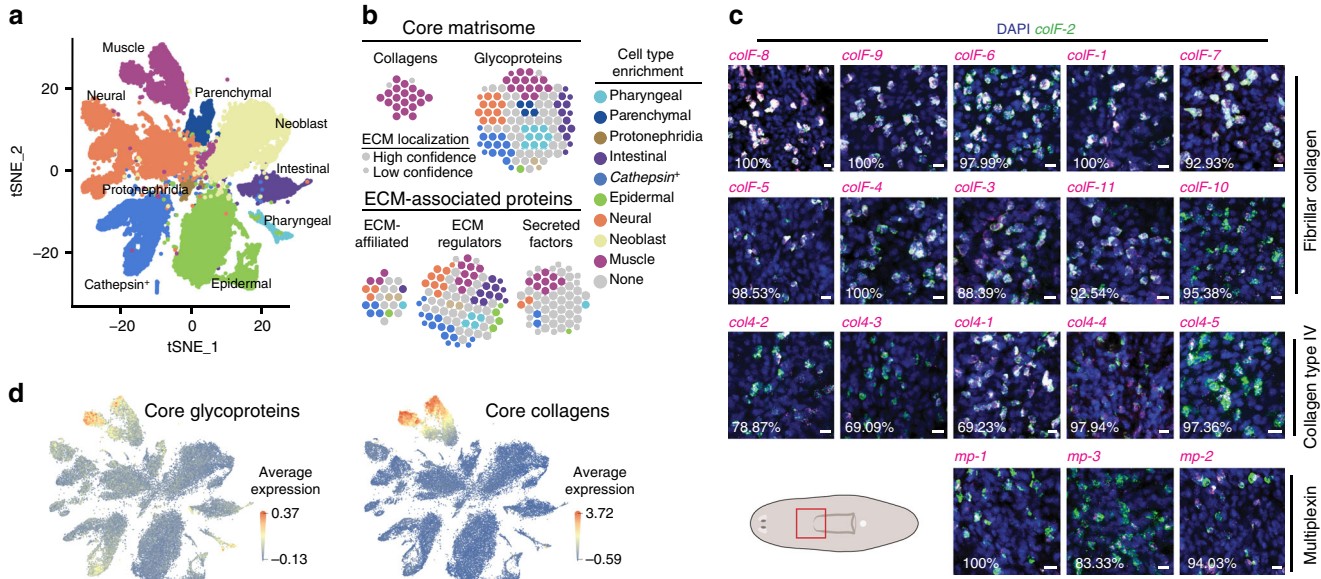

**Fig. 2** Collagen expression is specific to muscle cells. **a** tSNE dimensional reduction of gene expression in 50,562 cells[12] showing major cell types in planarians. **b** Circles represent a protein-coding gene within the matrisome, colored by most enriched cell type (greatest AUC) and sized by confidence in ECM localization. **c** FISH using RNA probes of indicated collagen-encoding gene (magenta) with percentage of cells double positive for the canonical planarian muscle marker *colF-2* (green). Red box in cartoon shows area imaged. Scale bars, 10 µm. **d** tSNE plot as in Fig. 2a with each cell colored and sized by the average normalized expression of high confidence core glycoprotein-encoding genes and collagen-encoding genes

including *colF-2⁻ mhc-1⁺* pharyngeal cells (Supplementary Fig. 5). *col4-5* is wound-induced in muscle cells around 12 h post amputation[41]. The planarian transcriptome also encodes three members of the multiplexin collagen family (mammalian type XV and XVIII collagens). These basement membrane proteins are characterized by a C-terminal endostatin domain, which can be cleaved off to form a signaling protein. *multiplexin-1 (mp-1)* had enriched expression in intestinal muscle and DV muscle[14,15]. We confirmed the specificity of *collagen* gene expression in muscle with whole-mount FISH and with analysis of 43 cell-type clusters and 142 cell-type subclusters from SCS data[12] (Fig. 2b, c, Supplementary Figs. 3–5).

**Planarian muscle is a major source of ECM**. Whereas collagen expression was clearly enriched in muscle cells, we wanted to determine if muscle cells are the major contributor to the ECM. Muscle cells expressed genes involved in protein secretion, but these genes did not display tissue-enriched expression (Supplementary Figs. 2 and 6, Supplementary Data 3). As an indirect proxy for total ECM secretion, we averaged the expression within a single cell of all core ECM components, each normalized by the mean expression of genes with similar expression levels[43]. Using this analysis for planarian PCGs, we recapitulated the finding that planarian muscle cells displayed enriched expression of genes encoding secreted and transmembrane patterning molecules, including Wnt, Bmp, Inhibin, Follistatin, sFRP, Notum, Noggin, Netrin, and Slit family proteins[7] (Supplementary Fig. 6).

The average expression of planarian core matrisome components, including collagens and the core glycoproteins, were strongly enriched in muscle cells, consistent with the hypothesis that muscle secretes the majority of ECM molecules (Fig. 2d). To further test whether muscle cells are the major source of ECM, we re-clustered the planarian SCS data using only the expression of high confidence core and ECM-affiliated matrisome components. The genes comprising the first principal component were all highly and specifically expressed in muscle cells (Fig. 3a, b). These genes encode 5/5 collagen type IV, 9/11

fibrillar collagens, the proposed extracellular collagen chaperone SPARC[27], plc[44], hmcn-1[45], and P4H4, an enzyme that post-translationally modifies proline residues to stabilize collagen structure[17]. Using the top six principal components of matrisome gene expression, one large (~10% of all cells) distinct cluster emerged (Fig. 3c). This cluster of 4952 cells consisted of 4726/5014 (94.2%) of all muscle cells and 226 (4.6%) non-muscle cells. Therefore, there is sufficient information within the small subset of gene expression space corresponding to matrisome gene expression to accurately identify muscle cells in SCS data, even blinded to the expression of the canonical actomyosin contractile machinery.

We analyzed a second independent Drop-Seq dataset (21,612 cells, with median 398 contigs detected per cell)[40] in addition to the dataset shown (50,562 cells with a median of 1,164 contigs detected per cell)[12] and a SMART-Seq2 SCS dataset, which had greater read depth per cell (413 cells, with median 4872 contigs detected per cell)[41]. We obtained similar results to those described above, supporting the conclusion that muscle is a major source of ECM (Supplementary Fig. 6). This conclusion is further supported by gene ontology (GO) analysis of SCS data, which identified 'basement membrane' as a GO term enriched in muscle cells[46]. Secretion of particular ECM components was not specifically confined to muscle cells (Fig. 2b, Supplementary Figs. 3, 4). Mucosal ECM components displayed enriched expression in various subsets of parenchymal cells, and intestinal cells expressed all three laminin subunits. However, the vast majority of expression of genes encoding collagens and a substantial amount of core glycoprotein expression occurred in muscle. We could not detect any other dedicated ECM-secreting cell in the complete planarian cell type atlas, suggesting that some of the connective-tissue functions supplied by cells such as fibroblasts in vertebrates are accomplished instead by muscle in planarians.

**Hmcn-1 maintains localization of multiple parenchymal cells**. To probe planarian ECM function, we inhibited muscle-specific

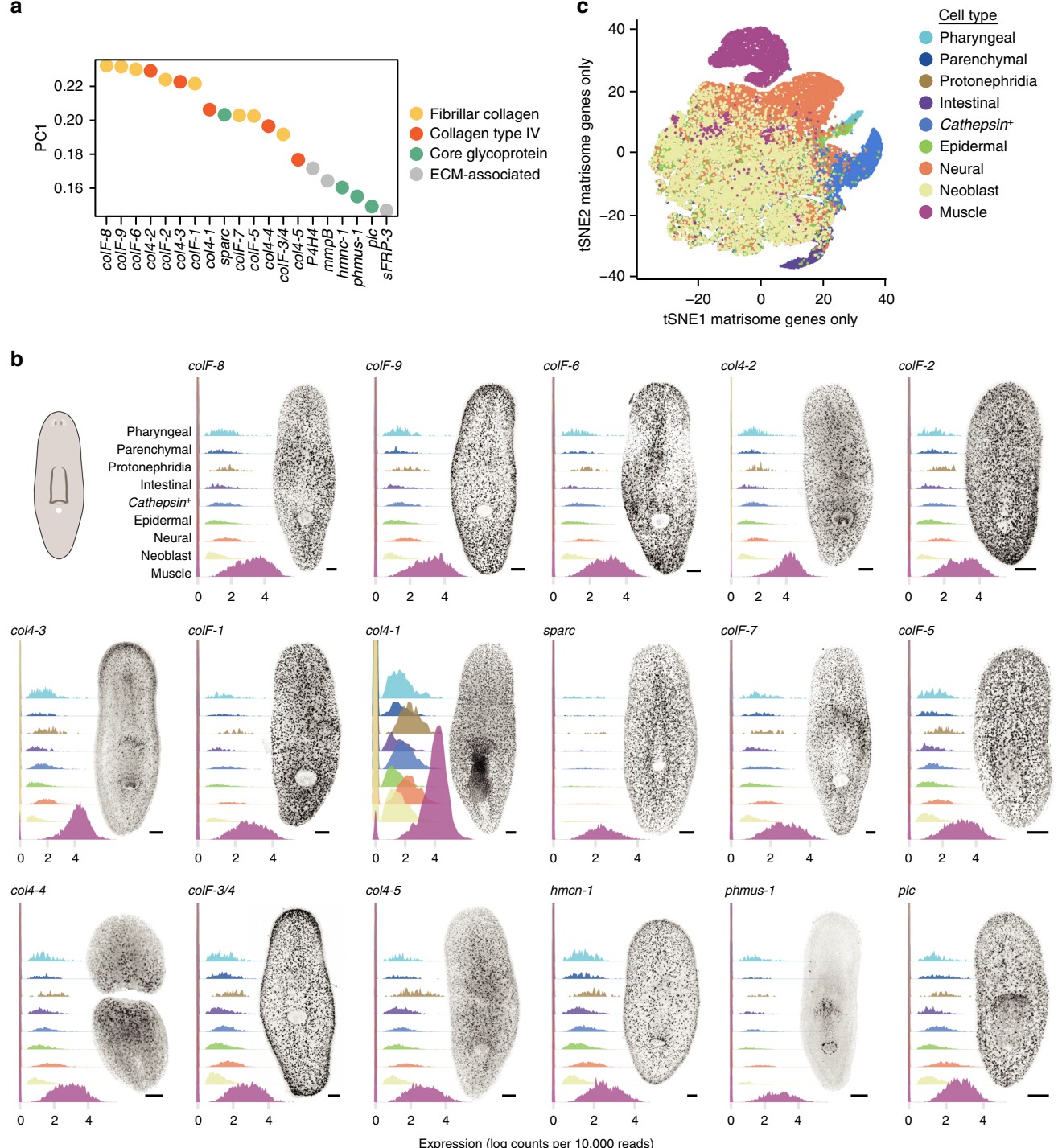

**Fig. 3** Muscle expresses collagens and core glycoproteins. **a** Principal component analysis using high confidence ECM components yielded PC1 loadings of indicated genes. **b** Ridge plots showing smoothed densities of indicated gene expression per cell (x-axis) within each cell type (colored). All y-axes are truncated equally to visualize densities of expression >0. Right, whole mount FISH of indicated gene with image representative of 4 animals. Scale bars, 100 μm. **c** tSNE representation of clustering using only high confidence ECM components, with each cell colored by its clustering in the full gene expression tSNE shown in Fig. 2a

genes encoding core glycoproteins or receptors predicted to interact with the ECM and found RNAi phenotypes affecting head tip and eye regeneration (*plc*), body wall muscle fiber elongation (*phred-1*[47]), and smoothness of the epidermal layer (*α-integrin-1*, EGF-repeat dd_10716, and *hmcn-1*) (Supplementary Fig. 7, 8, Supplementary Data 4). We further investigated the role of *hmcn-1*, which is most strongly expressed in the body wall

muscle (Fig. 4a, Supplementary Fig. 9a) and encodes a highly conserved core glycoprotein (Fig. 1c). Inhibition of *hmcn-1* during homeostatic tissue turnover led to severe epidermal ruffling and the ectopic supra-muscular localization of cells between 6G10+ muscle fibers and the sub-epidermal membrane, including mitotic (H3P+) neoblasts and *mag-1*+ gland cells (Fig. 4b–f, Supplementary Fig. 9b–d). Ectopic supra-muscular

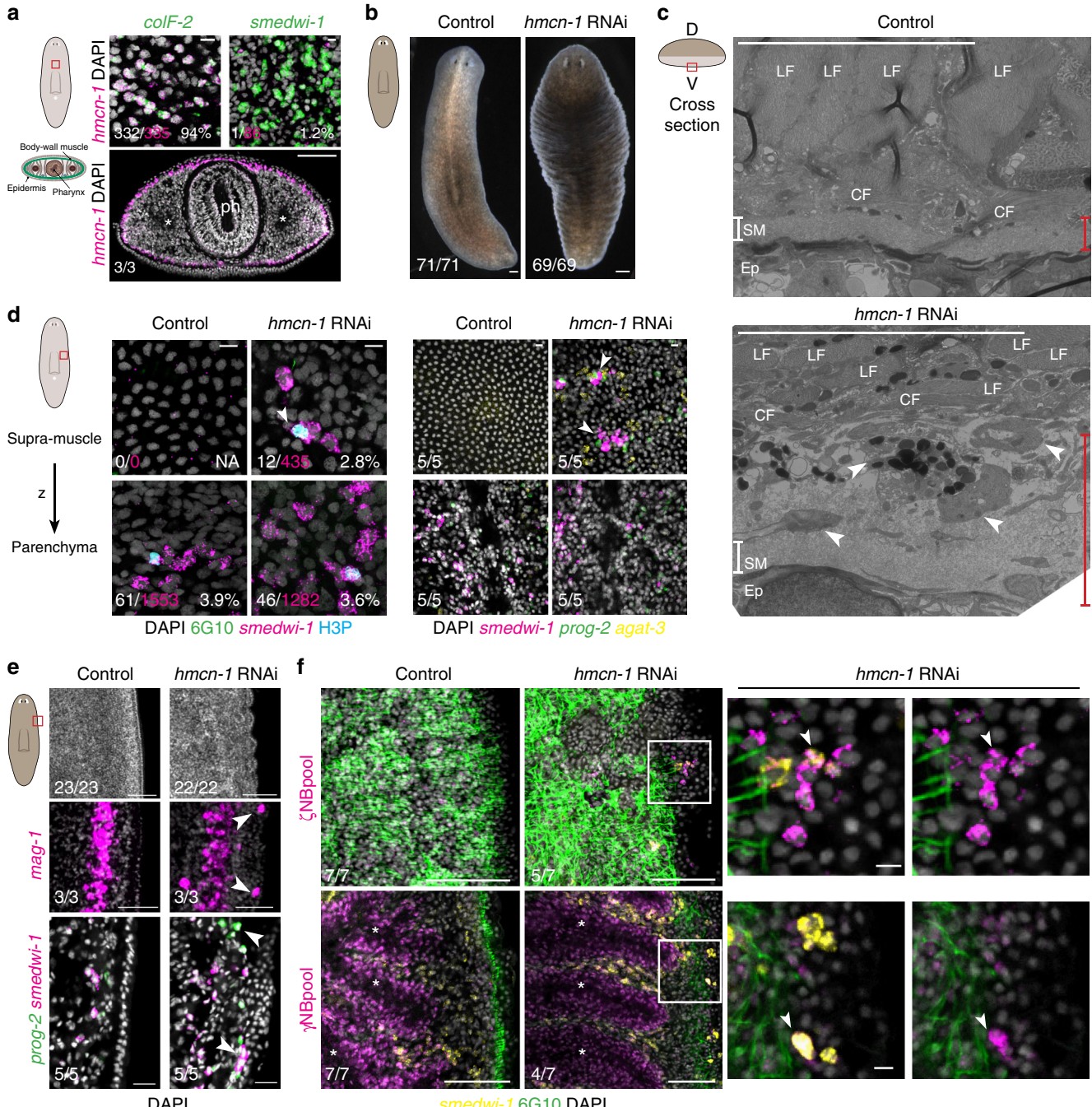

**Fig. 4** hemicentin-1 (hmcn-1) is required for maintaining parenchymal neoblast localization. **a** hmcn-1 expression is strongest in the colF-2⁺ sub-epidermal body wall muscle. Double-positive cell number of total hmcn-1⁺ cells from 1 representative of four animals. Ph, pharynx. Lower panel: scale bar, 100 μm. **b** Inhibition of hmcn-1 leads to severe epidermal ruffling in live animals. Scale bar, 200 μm. Four experiments. **c** TEM micrograph of ventral supra-muscle region representative of four sections from two animals. Red line indicates distance from basal epidermal surface (Ep) to circular muscle fiber (CF) layer. White line indicates area of high density ECM characterizing the sub-epidermal membrane (SM). Dorsal, up. Ep, epidermis. LF, longitudinal muscle fiber. **d** Dividing (H3P⁺) neoblasts (smedwi-1⁺) and cells of epidermal lineage (prog-2⁺, agat-3⁺) are found ectopically between the 6G10⁺ muscle fibers and the ventral epidermis. Left panel: total smedwi-1⁺ cells (magenta) outside or inside muscle fibers in the post-pharyngeal region counted in four animals. **e** Confocal imaging of different ectopic cells underneath the epidermis. Bottom panel shows cryosection. Scale bars, 100 μm. **f** Neoblast (NB) subpopulations were present outside of the 6G10⁺ muscle layer in hmcn-1(RNAi) animals. Right panels show white-boxed areas with and without smedwi-1 channel. Left panels, scale bar 100 μm. White arrowhead, ectopic cells. *indicates interior of an intestinal branch. Unless specified, 36 days RNAi, fractions indicate number of total animals from two independent RNAi experiments with phenotype shown in representative image. Anterior is up, RNA probes for FISH of indicated gene, scale bars, 10 μm

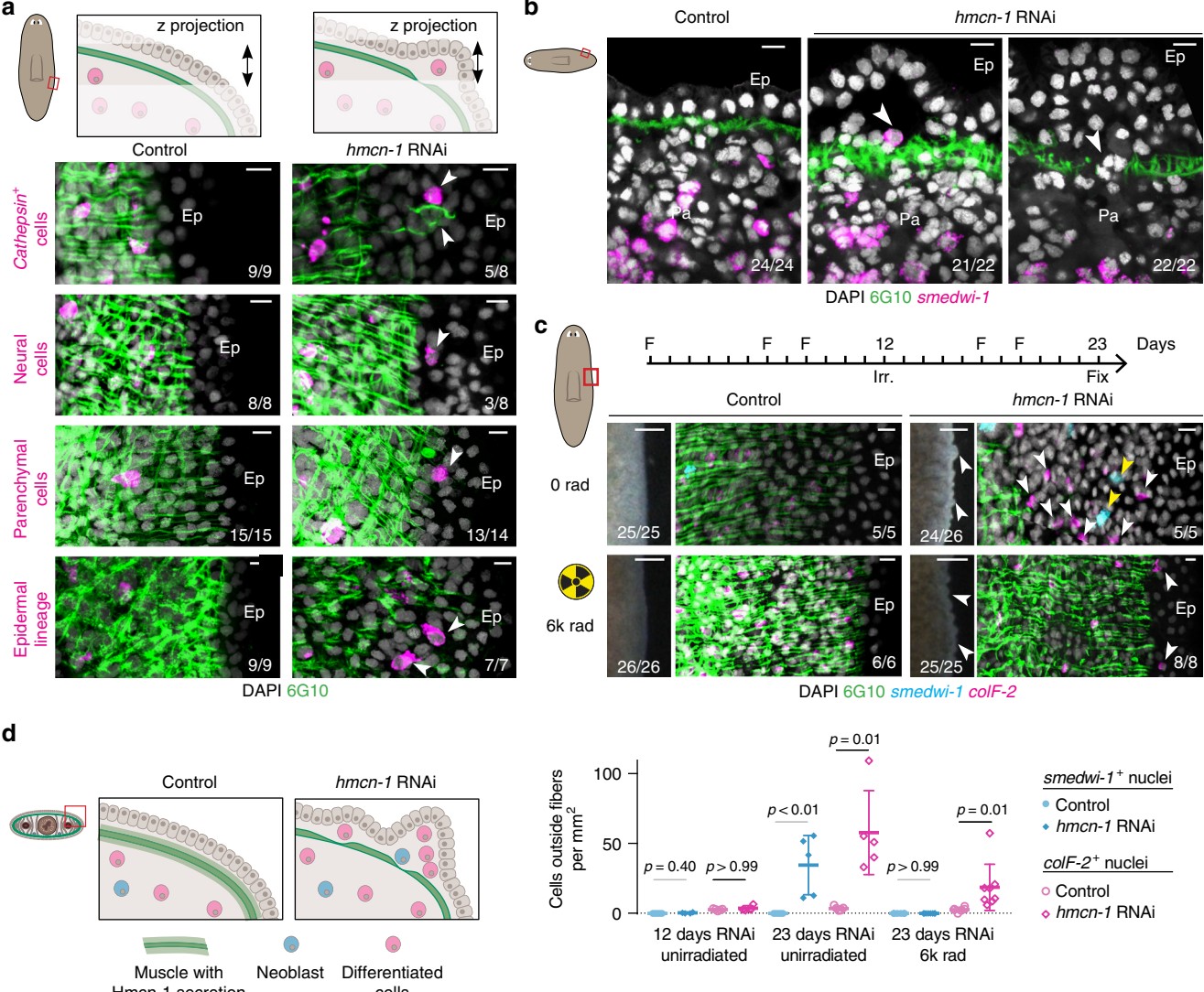

**Fig. 5** *hmcn-1* is required for maintaining parenchymal differentiated cell localization. **a** Confocal z-projections from the red rectangle as indicated in the diagrams (upper panel) show ectopic localization of cells expressing markers of differentiated cell types (Methods) outside of the 6G10⁺ muscle layer at 20 days of RNAi. **b** Confocal slice through side of the animal as diagrammed to the left shows ectopic localization of a *smedwi-1*⁺ neoblast and *smedwi-1*⁻ cells with non-epidermal nuclear morphology outside the parenchyma near a hole in the muscle layer at 20 days of RNAi. Anterior, left, four independent experiments. **c** Animals were fed with dsRNA (F) as indicated in top timeline and lethally irradiated at 12 days of RNAi. In the absence of neoblasts at 23 days of RNAi, epidermal ruffling was readily apparent (left image: scale bar, 100 μm) and *collagen*⁺ nuclei were found outside of the muscle layer, quantified in graph below. Number of nuclei outside of ventral muscle fibers normalized by animal area is plotted with mean, standard deviation, and *p*-value from Kruskal–Wallis test with Dunn's correction. **d** Diagram summarizing the defects after inhibition of *hmcn-1*. Ep, epidermis. Pa, parenchyma. White arrowhead, ectopic cells. Unless otherwise specified, fractions indicate number of total animals from 2 independent RNAi experiments with phenotype shown in representative image, anterior is up, scale bars, 10 μm

cells appeared to spatially recapitulate the inner-to-outer stages (*smedwi-1*⁺ neoblasts to *prog-2*⁺ to *agat-3*⁺ post-mitotic cells) of the planarian epidermal lineage[48,49] (Fig. 4d, Supplementary Fig. 9c), in a manner similar to cell colonies formed by single neoblasts in the parenchyma[50]. Ectopic specialized neoblasts of the epidermal (ζ) and intestinal (γ) lineages[51] were present (Fig. 4f). Occasionally small unpigmented growths appeared (Supplementary Fig. 9e), similar to ectopic cell phenotypes reported after inhibition of genes encoding MMPs[35,36] or tumor suppressors[52–55]. By contrast, in *hmcn-1(RNAi)* animals, clear defects were not apparent in the organization of the sub-epidermal membrane, mitotic neoblasts, or gut structure (Fig. 4c–f).

We next investigated the early *hmcn-1(RNAi)* phenotype at a time after RNAi initiation when animals showed minor epidermal

ruffling yet were similar to controls by H3P labeling and RNA sequencing (Supplementary Fig. 10, Supplementary Data 5). Sporadic differentiated cells and isolated neoblasts outside the muscle fibers were observed (Fig. 5a, b, Supplementary Movies 1 and 2). We used irradiation, which ablates all neoblasts, to determine whether ectopic neoblasts were solely giving rise to the ectopic cells or if post-mitotic differentiated cells were becoming ectopically localized. *hmcn-1(RNAi)* animals irradiated when supra-muscle cells were not yet present still developed a ruffled epidermis and ectopic localization of muscle and *cathepsin*⁺ cell nuclei (Fig. 5c, Supplementary Fig. 11). Ectopic differentiated cells were more abundant in unirradiated than in irradiated *hmcn-1(RNAi)* animals, suggesting additional neoblast-dependent contributions. The ectopic localization of both dividing neoblasts and differentiated cells following *hmcn-1* inhibition suggests that this

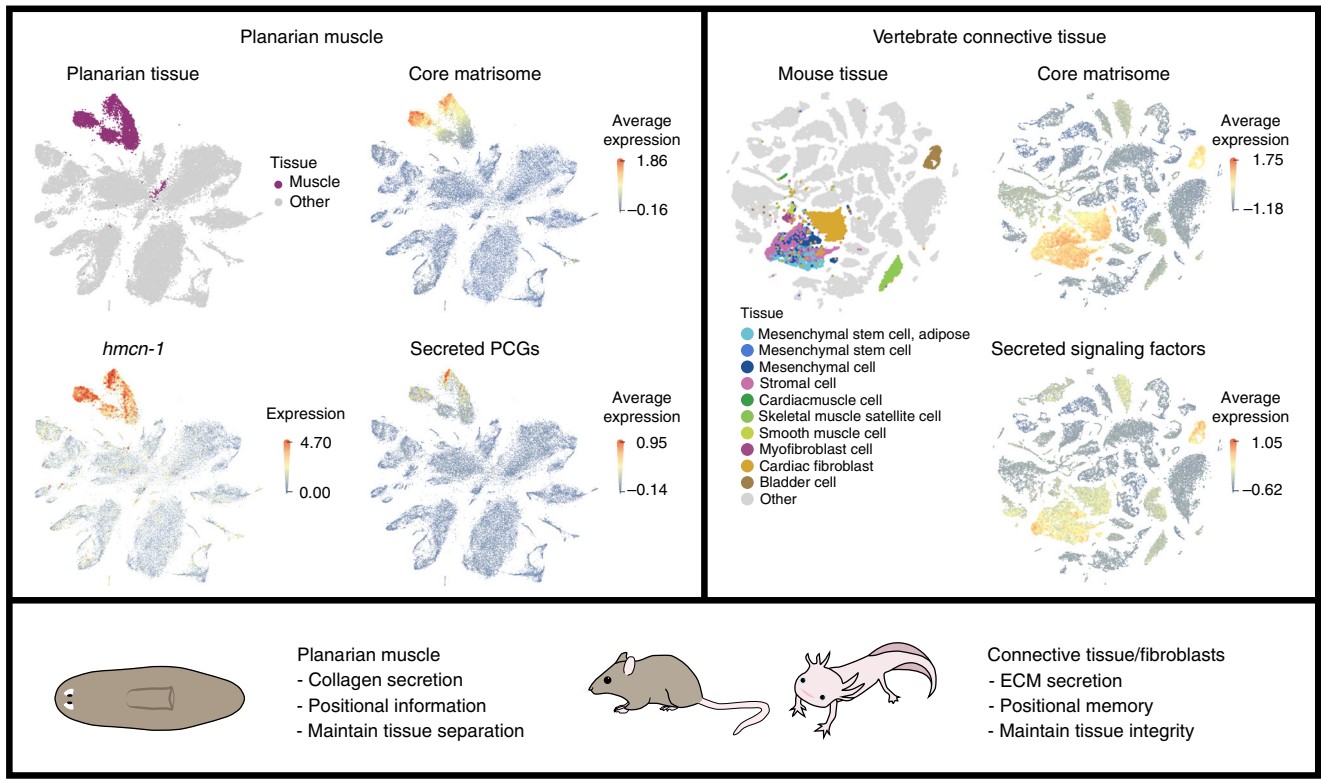

**Fig. 6** Functional similarities between planarian muscle and vertebrate connective tissue. Single cell sequencing from planarians and mouse are shown in two dimensional tSNE plots (top), highlighting cell populations of interest, as defined from previously published total gene expression data. The average expression of core matrisome, including *Smed-hmcn-1*, and secreted PCGs/signaling factors (Supplementary Data 3) show strong expression in the muscle (planarian SCS data[12], left) and connective tissue (mouse SCS data[57], right). Lower panel summarizes functional roles of planarian muscle and vertebrate connective tissues

muscle-derived glycoprotein broadly functions to maintain tissue separation of the inner parenchyma and outer epidermis (Fig. 5d).

**Matrisome expression data in mouse SCS data**. We sought to compare our analysis of matrisome expression in planarian SCS data to data from other organisms. Recent single-cell RNA sequencing efforts have defined the transcriptomes of cell types in multiple different organisms, including multiple mouse cell types as an example vertebrate[56,57]. Combining these SCS data with the published set of matrisome components[24], we performed similar average expression analysis to approximate levels of murine ECM secretion. Fibroblasts in stromal tissues are a predominant source of ECM, and chondrocytes and osteoblasts form specialized ECM structures such as cartilage and bone[26]. This well-known predominance of ECM secretion in stromal cells and the mesodermal stem cells from which they are derived was readily apparent by analyzing core matrisome gene expression in mouse SCS data (Fig. 6). The set of ECM components conserved in both species was expressed in mouse stromal and planarian muscle cells (Supplementary Fig. 12). In addition to examining the core matrisome components in mouse data, we also assayed the expression of various signaling molecules, including homologs of planarian PCGs (Supplementary Figs. 13–15). Secreted signaling molecules have a myriad of roles during development and adult homeostasis in multiple tissues. Whereas Wnt ligands control division and differentiation in multiple contexts, the high expression of Wnt ligands is found, as expected, in cells isolated from the adult epidermis consistent with the key roles of Wnt signaling in epidermal and hair follicle homeostasis[58]. Connective tissues and the mesenchymal stem cells that give rise to the connective tissues, muscle, and

immune cells showed high expression of TGF-beta, FGF, and sFRP-family secreted ligands. Connective tissue was among the murine tissues that showed abundant average expression of all secreted signaling factors (Fig. 6).

## Discussion

Connective tissues and ECM surround cells and tissues to stabilize, shape, and support their function during development, adult homeostasis, and regeneration[18–20]. Here, we identified ~250 genes predicted to encode proteins that comprise the planarian matrisome and, utilizing three independent SCS datasets[12,40,41], identified muscle as a major source of planarian ECM. The planarian matrisome will aid in understanding the evolution of ECM components, and could be expanded in the future using biochemical and bioinformatics approaches to identify any additional ECM proteins with planarian-, spiralian-, or protostome-specific protein domains.

We utilized a complete planarian cell type-transcriptome atlas to map the expression of all predicted matrisome genes onto all adult cell types. Within the planarian matrisome defined in this work were 19 collagen-encoding genes, and all of these genes displayed enriched expression in muscle. These findings indicate that there is lack of a distinct non-muscle planarian cell type with ECM secretion as a major function, similar to vertebrate fibroblasts. Instead we propose this ECM-secreting connective tissue function is largely performed by planarian muscle.

Connective tissues are diverse in their structure and composition, however, they are similar in providing structural support and are associated with abundant ECM secretion. We found the expected strong enrichment of matrisome gene expression in various mouse stromal cells in single-cell sequencing datasets,

although multiple different immunological, contractile, epidermal, and endothelial cell types are also known to secrete ECM. Fibroblasts, a major cellular component of connective tissues, perform multiple functions in addition to secreting ECM. Fibroblasts migrate to wounds, proliferate, activate to become contractile myofibroblasts, can transform into different cell types, and can influence stem cells during regeneration[25,26,59]. Neoblasts are the only dividing planarian somatic cells and are responsible for all new tissue production in regeneration. Therefore, planarians lack an apparent fibroblast-like cell type that has all functional attributes of fibroblasts. We therefore considered ECM formation as the main point of comparison for defining a planarian cell type that provides a structural, connective tissue-like, function.

Our proposal that planarian muscle has connective tissue-like function is further supported by the disruption of tissue organization caused by inhibition of the highly conserved core glycoprotein *hmcn-1*. In *hmcn-1(RNAi)* animals, neoblasts and other differentiated tissues can become ectopically localized peripheral to muscle, juxtaposed to the sub-epidermal membrane. This phenotype is similar to blistering between the basement membrane and dermis seen during fin development in zebrafish *hmcn-1* mutants[60]. In *C. elegans*, the *hmcn-1* homolog HIM-4[45] and the integrin INA-1/PAT-3 help anchor basement membranes to each other[61]. The ectopic cell localization phenotypes of planarian *hmcn-1* and *inta-1* would be consistent with a similar complex helping maintain parenchymal cell localization in planarians. Surprisingly, ectopic cells in *hmcn-1(RNAi)* animals appear largely normal. For instance, the cells that ectopically localize to the supra-muscular space show proper expression of cell-type-specific marker genes, ectopic muscle cells had long processes typical of muscle cells, and ectopic neoblasts divided and specialized as per their normal behavior. The *hmcn-1* phenotype offers the possibility to investigate the competency of different cell types, including neoblasts, to properly function outside of the parenchyma.

Positional information refers to factors that influence the regional identity of cells. Understanding the process of regeneration requires understanding how positional information is encoded[62–64]. Muscle has a prominent role in maintaining and regenerating positional information in planarians. Both pre-existing muscle and newly specified muscle change expression during different phases of the wound and regenerative responses[1,2,7,13,41]. Specifically, planarian muscle is the primary site of expression of patterning molecules (e.g., Wnt and Bmp ligands), encoded by PCGs, which control regeneration outcomes and maintain proper regional tissue identity[7]. These molecules are constitutively expressed in a regionalized manner in adult planarian muscle, where they regulate tissue identity and pattern during homeostatic cell turnover[1]. The ablation of multiple muscle subtypes leads to aberrant patterning, and RNAi conditions that result in failure to reestablish PCG expression domains after wounding also result in failed regeneration[14,65]. The co-incidence of enriched PCG expression and ECM expression in the same tissue – muscle – described here suggests that muscle generates major components of the supporting extracellular environment for dividing neoblasts and other parenchymal cells and also numerous signaling molecules existing in this extracellular environment.

Multiple lines of evidence suggest that vertebrate connective tissues, including fibroblasts, display positional variation and potentially influence regional tissue identity during appendage regeneration. For example, human skin fibroblasts can maintain regional Hox gene expression, indicative of different positional identities[66], different dermal mouse fibroblast populations participate in wound healing[67], and proximal-distal differences in

fibroblasts are maintained during mouse digit tip regeneration[68]. In zebrafish fin rays, constitutive differential transcription factor expression occurs in fibroblasts and osteoblasts, and these factors can modulate anterior–posterior variations in bone morphology during regeneration[69]. Finally, in axolotls, connective tissues display positional memory during limb regeneration, whereas Schwann cells and muscle cells do not[70–72], and connective tissue can re-express genes that control regional limb development in a manner suggestive of positional identity[73]. Based on these observations, we suggest that planarian muscle and vertebrate connective tissue may be functionally similar regarding the production of ECM to support tissue architecture and potentially by informing adult positional identity during regeneration.

These specific functional similarities of planarian muscle cells and vertebrate connective tissue are of interest for understanding their vital roles in adult tissue biology and the evolution of specialized cell types involved in adult positional identity and regeneration. For instance, it will be of interest to determine whether ECM secretion is an attribute found broadly in cells important for harboring positional information across metazoans. Future studies across multiple animal clades could support a model in which these specialized roles for planarian muscle and vertebrate stroma cells evolved from a common ancestral cell state or could indicate that these functional similarities reflect convergent evolution. For example, further taxon sampling might suggest that the last common ancestor of planarians and vertebrates possessed a specific mesodermal cell type with contractile, secretory, and patterning functions. If so, one potential trajectory could have involved the emergence of multiple specialized cell types from this ancestral state, with partitioning of roles into different cells – for instance, fibroblasts in vertebrates could have emerged from this ancestral state with roles in ECM secretion and maintaining positional identity. Distinguishing between these and other possible models for the evolution of adult positional information and connective tissue might offer clues as to which cell types are broadly crucial for regenerating and maintaining adult patterning. We conclude that planarian muscle acts as a connective tissue for maintaining tissue architecture through ECM secretion, suggesting functional similarities between cells associated with adult patterning information in vertebrates and planarians.

## Methods

**Matrisome categorization and nomenclature.** Interpro domain identifiers[24] from [http://matrisome.org] were used to query PlanMine[30,31] [http://planmine.mpi-cbg.de] dd_Smed_v6 with $p < 0.1$ using the python API intermine v1.11.0 and collated using a custom python script using python 3.6.4 with pandas v0.22.0 and numpy v1.14.0. Contig ids with at least one matrisome defining domain were categorized as Core matrisome, ECM affiliated, ECM regulators, Secreted Factors, were kept for further analysis and tentatively categorized based on the maximum number of domains present in each category. Contigs were removed from further consideration in a respective category if they contained an excluding domain of that category [http://matrisome.org] . Most excluding domains were kinase, phosphatase, integrin, or 7tm domains. Manual inspection revealed that some excluding domains were filtering out ECM proteins (i.e. TIL domain in dd_2649 encoding a mucin-like protein would exclude it from the core matrisome), therefore the following domains were not considered as excluding domains for any matrisome category in planarians: IPR001073 (C1q), IPR002007 (An_peroxidase), IPR002919 (TIL), IPR020067 (FRI), IPR018933 (C345C), and IPR002350 (Kazal). Presence of these domains were used subsequently to categorize putative matrisome proteins.

Reciprocal blastx/tblastn between human RefSeq proteins (downloaded 3/7/2018) and the dd_v4 transcriptome assembly was performed using -max_target_seqs 1. Contigs from dd_Smed_v4 with either blastx or tblastn best blast hit ($e$-value <0.01) to any of the RefSeq proteins sequences (RefSeq ids or RefSeq ids associated with matrisome gene names) were tentatively categorized based on the ECM categorization of the human blastx hit. Contigs with either a blast-based matrisome categorization or an interpro-based matrisome categorization were kept for further analysis. Combining this list with the list of contigs with a putative ECM domain yielded 767 contigs (Supplementary Data 1, 2).

The longest ORF for each contig based on AUGUSTUS v3.3 gene predictions[31], CLUSTEL-omega alignments of homologous sequences from multiple planarian transcriptomes, genomic sequence, read density, and co-expression in single-cell data was determined. If an ORF consisted of multiple contigs, the contig with the largest average read count in the single-cell sequencing data was used as the representative contig for the protein for subsequent expression analyses. Contigs which are part of the same gene are listed at the bottom of Supplementary Tabel 1. Contigs containing one or more TM domains (tmhmm[74]), with care was taken not to count a potential signal peptide as a TM domain, were removed (Supplementary Data 2) while known transmembrane ECM components (collagen, MMP) were kept as matrisome components. SignalP v4.0 with high sensitivity (-u 0.34 -c 120) was used to determine presence of a signal peptide. predGPI was used to predict presence of a C-terminal GPI anchor which was used to categorize proteins as ECM-affiliated. Differences between domain-based categorization and blast-based categorization were resolved manually based on domain structure organization. Proteins with reciprocal best blast hits to human matrisome proteins were inspected for domain architecture similarities using SMART with HMMER searches of Outlier homologs, PFAM domains, and signal peptide prediction. Genes encoding reciprocal best blast hits with similar domain organization were named after the human homolog. Collagens were named colF-1 through colF-11, col4–1 through col4–5, and multiplexin-1 (mp-1) through mp-3 as invertebrate collagen nomenclature is not standardized.

Matrisome proteins were additionally categorized high and low confidence of extracellular matrix localization. High confidence proteins include all proteins with clear domain organization homologous to a well-established matrisome component. Additionally, contigs with clear signal peptide with definite matrisome-defining domains, even with unclear homology by domain organization, were taken as high confidence matrisome components. Contigs with best blast hits to non-matrisome proteins with plasma membrane localization (e.g., mannose receptor) but which did not encode characteristics of a plasma membrane associated protein (no TM, no clear intracellular domains) were also taken as high confidence matrisome components. Low confidence matrisome components include those which have a signal peptide but no clear matrisome-defining domain. Additionally, genes which likely encode components present within the secretory pathway, such as the lysosomal cathepsin proteases, cubulin, proteins with only Ig repeats, or proteins likely involved in lipid transport. The full cds of each protein sequence was re-analyzed with interproscan v5.31 with redundant domain annotations removed to obtain a list of non-overlapping domain annotations, maintaining the annotation with the lowest p-value. Graphing of matrisome component selection and cell-type specific values in Figs. 1a and 2b was performed using RAWGraphs v1.0.0 at [https://rawgraphs.io]. The resulting in silico matrisome gene names, protein sequences, domain annotations, categorization, and single cell sequencing cluster with highest AUC value are listed in Supplementary Data 1. In addition, gene encoding proteins containing putative matrisome domains or blast hits to matrisome proteins which were removed from the matrisome list (cell-surface receptors with transmembrane domain(s), proteins without signal peptides, and gene with unclear transcripts) are similarly listed in Supplementary Data 2.

**Single-cell sequencing data analysis.** Read count/UMI count, tissue of origin, cell ontology or cell-type assignments, and tSNE embeddings were stored as Seurat v2.3 objects and analyzed in R 3.4.4. Average expression was calculated and normalized for indicated gene set using SetModuleScore function[43] (Seurat). Average gene set expression or single gene expression (counts or UMIs per 10,000 reads) was visualized using a customized FeaturePlot function with both color and size proportional to expression value plotted in ascending order of expression. Ridgeplots were created using ggridges v0.4.1 for calculating joint bandwidth for all cell types and plotting the densities. All ridgeplots were cropped uniformly to allow for visualization of expression densities without the overpowering peak at 0 counts. For reclustering using matrisome gene expression, clustering and tSNE embeddings were performed using only the high confidence matrisome contigs using the top 6 pca components of 20 calculated with tsne perplexity 40. AUC values of gene module scores (0.5 categorization at random chance, 1.0 perfect categorization) were calculated with AUC-package v0.3.0.

**Gene set enrichment analysis.** Average log fold changes between cells within broad cell-type clusters and all other cells were calculated as in the Seurat Find-Markers function. Genes expressed in >1% of cells at 0.5 were ranked by the average log fold change before using fgsea v1.4.1 with gsea parameter of 1.0 to determine gene set enrichment statistics. Gene sets for myogenesis and protein secretion are reciprocal best blast hits between dd_Smed_v6 and human proteins, as uniprot ids, listed in M5909 and M5910 Hallmark GSEA gene sets available at [http://software.broadinstitute.org/gsea/msigdb/genesets.jsp?collection=H]. Gene sets are listed in Supplementary Datas 1, 3.

**Animals.** Asexual Schmidtea mediterranea strain (CIW4) animals were fed calf liver and starved 5–7 days prior to use in experiments.

**Cloning and RNA probes.** Nested primers with adapters were used to amplify genes from planarian cDNA, as listed in Supplementary Data 6. For cloning collagen sequences, the highest molecular weight band was isolated using MinElute Gel Extraction Kit (Qiagen). Genes were cloned into pGEM (Promega) and confirmed by Sanger sequencing (Genewiz). Sixteen microlitre of PCR product using the adapter sequences were used to generate RNA probes with Dig (Roche 11277073910), DNP (Perkin Elmer NEL555001EA), or Fitc (Roche 11685619910) modified nucleotides in an overnight T7 reaction (Promega P2077). Probes were ethanol precipitated and resuspended in 50 μl formamide. For cell type probe pools, RNA probes for multiple specific cell subtypes[12] were synthesized separately using the same hapten and each used at 1:800 during hybridization. Gamma-Neoblast pool: hnf4 (dd_Smed_v4_1694_0_1), prox-1 (dd_Smed_v4_13772_0_1), and gata4/5/6 (dd_Smed_v4_4075_0_1). Zeta-Neoblast pool: zfp1 (dd_Smed_v4_18207_0_1) and soxP-3 (dd_Smed_v4_5942_0_1). cathepsin+ cell probe pool: dd_Smed_v4_1103_0_1 (AQP1), dd_Smed_v4_10872_0_1 (PTPRT). Neural cell probe pool: dd_Smed_v4_6208_0_1 (ChAT), dd_Smed_v4_12700_0_1 (sert), dd_Smed_v4_8060_0_1, dd_Smed_v4_3814_1_1 (ITPR3). Parenchymal cell probe pool: dd_Smed_v4_515_0_1, dd_Smed_v4_43_0_1, dd_Smed_v4_628_0_1 (SSPO), dd_Smed_v4_4761_0_1 (ANO7), dd_Smed_v4_238_1_1 (ZAN6). Epidermal lineage probe pool: dd_Smed_v4_69_0_1 (prog-2), and dd_Smed_v4_2385_0_1 (agat-3).

**Fluorescence in-situ hybridization and immunostainings.** Animals were fixed and processed for in situ hybridization using tyramide signal amplification[11]. In brief, animals fixed using 5% NAC in PBS for 5 min and 4% formaldehyde in PBST (PBS with 0.3% Triton X-100) for 20 min followed by dehydration and storage at −20 °C in methanol were rehydrated, bleached in formamide bleach solution for 1–1.5 h, incubated in 2 μg/mL Proteinase K (Roche) for 10 min, and post-fixed in 4% formaldehyde before 2 h incubation at 56 °C in pre-hybe (hybe solution without dextran sulfate) and overnight incubation with RNA probes in hybe solution (4X SSC, 1% Tween-20, 1 mg/mL yeast RNA, 5% dextran sulfate, in 50% de-ionized formamide (Ambion)). After desalting at 56 °C through pre-hybe, 50% pre-hybe 50% 2X SSC, 2X SSC, and 0.2X SSC for 30 min each, animals were immunostained serially. All immunostainings were blocked for at least 30 min room temperature prior to overnight incubation at 4 °C in 5% Western Blocking Reagent (Roche 11921673001) and 5% heat-inactivated horse serum, washed 6 times 20 min each in PBST before 10 min RT tyramide development in TSA solution (0.003% $H_2O_2$ 20 μg/mL 4IPBA in 100 mM Boric Acid, 2 M NaCl pH 8.5). Quenching of HRP using 1% sodium azide 1.5 h RT was followed by six PBST washes. HRP-conjugated antibodies were used at the following concentrations: anti-Dig-POD (Roche 11633716001) 1:1000 with rhodamine tyramide 1:1500, anti-Fitc-POD (Roche 11426346910) 1:2000 with Cy5 tyramide 1:500, and anti-DNP-HRP (Perkin Elmer FP1129) 1:100–1:200 with 1:2000 fitc tyramide. Anti-muscle mouse monoclonal antibody 6G10[75] (DSHB 6G10–2C7) was used in a 1:500 dilution or VC1 (gift from K. Agata) was used at 1:5000 followed by Alexa-488, Alexa-568, or Alexa-647 conjugated anti-mouse IgG (H+L) antibody (Invitrogen A-10029, A-11031, and A-21235) at a 1:500 dilution. Anti-H3P (Millipore 05–817R-I, clone 63–1C-8) was used at 1:300 followed by anti-rabbit-HRP (Thermo Fisher 65–6120) 1:300 and developed with Fitc-tyramide 1:3000 in PBST + 0.003% $H_2O_2$. V5277 rabbit polyclonal[14] was used at 1:500 followed by Alexa-488 conjugated anti-rabbit IgG (H+L) antibody (Invitrogen A-11034) on animals fixed with 2% HCl (30 s RT) and Carnoy's fixation (2 h on ice).

**Imaging.** Animals were mounted ventral up in Vectashield mounting medium. Fluorescent images were taken with a Zeiss LSM700 Confocal Microscope using ZEN 2011 software with following objectives. For transverse cryosection shown in Fig. 4e and Supplementary Fig. 9b, stained animals were embedded in Tissue-Tek O.C.T. Compound (Electron Microscopy Services) and sectioned at 25 μm. All images and quantification for co-expression and for determining cell localization relative to muscle fiber layer were analyzed at 40X NA 0.75 with minimum $0.31 \times 0.31 \times 1.11$ μm resolution or 63X NA 1.4 with $0.2 \times 0.2 \times 0.57$ μm resolution with the exception of H3P+smedwi-1+ (Fig. 4d) which was quantified at 20X NA 0.8 with minimum $0.63 \times 0.63 \times 2.04$ μm resolution. Animal overview images were acquired with 10X NA 1.45 with minimum $1.25 \times 1.25 \times 7.26$ μm resolution. Linear adjustments of brightness and contrast equally across controls and experimental images, re-assignment of linear look-up table colors, co-localization analysis of FISH signals, and maximum intensity projections through area of interest was performed using Fiji with ImageJ 2.0.0[76]. Live animal images were taken with a Zeiss Discovery Microscope using AxioVision v4.7.1.

**RNAi.** dsRNA was prepared from in vitro transcription reactions (Promega P2077) using PCR-generated templates with flanking T7 promoters. Forward and reverse reactions were combined for ethanol precipitation and annealed after resuspension in water (48 μl water per initial 32 μl forward or reverse strand PCR product). 10 μl dsRNA was mixed with 30 μl liver and 2 μl per animal was used in RNAi feedings every 3–4 days or as described. 20 days of RNAi corresponds to 4 feedings and fixation 5 days post amputation (dpa). Thirty-three to 36 days RNAi corresponds to nine feedings (eight feedings for plc RNAi) during tissue homeostasis. All control animals were fed dsRNA for C. elegans unc-22 sequence not present in the planarian genome.

**Statistical testing**. Number of nuclei outside of muscle fibers per mm$^2$ (Fig. 5c) were not normally distributed (D'Agostino and Pearson normality test on all data of a particular cell type) and were compared using the nonparametric Kruskal–Wallis test with pairwise comparisons as shown and Dunn's correction for multiple hypothesis testing. Number of H3P$^+$ cells were compared using a two-sided Student's $t$-test (Supplementary Fig. 10c). Eye-to-head tip distance (Supplementary Fig. 7b) of single animals amputated into head and trunk pieces were analyzed using a two-way ANOVA with Sidak's multiple comparison test of RNAi conditions at each time point. Animals without data at 10 dpa (control animal 6, *plc* (RNAi) animals 1, 5) were excluded from analysis. All statistical testing on animal data was performed with GraphPad Prism v7.0. Images were blinded for counting *collegen* co-expression.

**mRNA sequencing**. Total RNA was isolated using Trizol (Life Technologies) from 5 pooled tails (0–5 min post amputation) at 13 days of RNAi in triplicate. Libraries from 1.0 µg of total RNA were prepared using Kapa Stranded mRNA-Seq Kit Illumina Platform (Kapa Biosystems). An additional 0.7X Ampure bead clean-up was performed to remove adapter dimer before 11 PCR cycles of library amplification. Libraries were sequenced on Illumina Hi-Seq and mapped to the dd_Smed_v6 transcriptome[30] using bowtie v1.2.2[77] with -best alignment parameter. Reads from the same isotig (differing only by final digits) were compiled using samtools v1.9 and summed to generate raw read counts for each contig. Differential expression analysis was performed using DESeq2 v1.18.1[78]. Average log$_2$ fold-change between *hmcn-1* RNAi and control were determined for sets of cell-type enriched genes (AUC >0.80)[41].

**Transmission electron microscopy**. Animals were kept on ice for 10 min before fixation with cold 2.5% glutaraldehyde, 3% paraformaldehyde with 5% sucrose in 0.1 M sodium cacodylate buffer (pH 7.4) overnight, then post-fixed in 1% OsO$_4$ in veronal-acetate buffer. Animals were stained overnight with 0.5% uranyl acetate in veronal-acetate buffer (pH 6.0), dehydrated, and embedded in Spurr's resin. Sections were cut on a Reichert Ultracut E microtome with a Diatome diamond knife at a thickness setting of 50 nm then stained with 2% uranyl acetate and lead citrate. The sections were examined using a FEI Tecnai spirit at 80 KV and photographed with an AMT CCD camera.

**Reporting Summary**. Further information on experimental design is available in the Nature Research Reporting Summary linked to this article.

## Data availability
The authors declare that all data supporting the findings of this study are available within the article and its supplementary information files or from the corresponding author upon reasonable request.

All code and raw imaging data is available upon request. Genes cloned and named in this study are deposited under Genbank MK430143 - MK430176. RNA-sequencing dataset generated by this study have been deposited in the NCBI GEO database under accession code GSE119840.

Planarian single-cell sequencing data were obtained from [https://shiny.mdc-berlin.de/psca/][40], from O. Wurtzel and is available at [https://radiant.wi.mit.edu/app/] and SRA: PRJNA276084[41], and from C.T. Fincher and is available at [https://digiworm.wi.mit.edu] and GSE111764 (ref[12]). The mca data[56] were obtained from [https://satijalab.org/seurat/mca.html] and re-embedding using FIt-SNE. The Tabula muris data[57] was obtained from [https://github.com/czbiohub/tabula-muris]. Murine matrisome data was obtained from [https://matrisome.org].

Source data underlying Fig. 5c, Supplementary Fig. 7b, and Supplementary Fig. 10c are available as a Source Data file.

A reporting summary for this Article is available as a Supplementary Information file.

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

## Acknowledgements

We thank N. Watson for TEM, C.T. Fincher for pre-publication access to SCS data, K.D. Atabay for cryosectioning, I. Oderberg for illustrations, and C. McMann for quantifying co-expression. We are grateful to the members of Reddien lab and R.O. Hynes for helpful comments on the project and manuscript. P.W.R. is an Investigator of the Howard Hughes Medical Institute and an associate member of the Broad Institute of Harvard and MIT. We thank the Eleanor Schwartz Charitable Foundation for support.

## Author contributions

L.E.C. and P.W.R. designed experiments and wrote manuscript. L.E.C. performed bioinformatic analyses and evaluated RNAi phenotypes. E.S. aided with FISH and RNAi experiments.

## Additional information

**Competing interests:** The authors declare no competing interests.

