## [Peer Review File · Nature Communications]

Reviewers' Comments:

Reviewer #1:

Remarks to the Author:

In this manuscript entitled "Muscle functions as a connective tissue in planarians", Cote and collaborators identify muscle, which convey positional information crucial for proper tissue regeneration, as the major ECM-producing cell type in planarians. To do so, the authors first defined the "matrisome" of planarians using a bioinformatic workflow based on protein sequence analysis that was originally devised to predict the human and mouse matrisomes. They identified 258 genes encoding core matrisome or matrisome-associated proteins and further classified them in different structural/functional categories. With this tool in place, the authors next sought to identify which cell type(s) express matrisome genes in planarians. Using previously published single-cell transcriptomic data, the authors showed that the main cell type expressing matrisome genes are muscle cells. They thus propose that the muscle is the connective tissue compartment in planarians. Last, the functional relevance of muscle-produced ECM components in tissue regeneration is illustrated through the example of the hemicentin-1 (*hmcn-1*) gene, encoding a conserved core matrisome glycoprotein. The authors demonstrate that *hmcn-1* knockdown results in the mislocalization of several cell populations including neoblasts, and epidermal and non-epidermal cells, indicating that muscle-produced ECM components can serve as cues to maintain proper tissular organization.

This is a very clearly written manuscript that reports not only the development of a novel resource but also proposes the new and exciting concept that the planarian muscle is a connective, ECM-producing, tissue. Furthermore, it illustrates how the in-silico definition of the planarian matrisome can assist with big data annotation and the identification of novel functionally relevant genes and proteins. This publication will have without a doubt a significant impact and can appeal to a very broad readership beyond planarian aficionados, including researchers interested in studying the evolution of the ECM and ECM proteins, ECM biologists, and researchers interested in tissue regeneration.

As my expertise lies in ECM biology and per the editor's request, my few minor comments focus on the first part of the paper.

1) Is there a reason to suspect that other ECM proteins that do not have structurally related mammalian orthologs may be encoded by planarian genes? The example that comes to mind is the cuticular extracellular matrix of nematodes: it is an ECM that does not have a mammalian equivalent and the components of this type of ECM (cuticlins, chitins) cannot be identified using the canonical "ECM domains" originally used to define mammalian matrisomes. Can the authors comment on that?

2) Page 5: Can the authors indicate how many proteins were removed by applying the size constraint of 100 amino acids. Several mammalian matrisome components, in particular belonging to the matrisome-associated division and secreted factors category are smaller than 100 amino acids (a family that comes to mind is the S100 family), it would thus be interesting to briefly discuss why this size constraint was applied.

3) Page 5 and Method section: the authors indicate using the presence of a transmembrane domain as a method to eliminate non-matrisome proteins. However, in the mammalian matrisome, several matrisome proteins including collagens, proteoglycans (syndecans, glypicans), and proteases (MMPs) are transmembrane proteins. Were only proteins displaying 7-transmembrane domains eliminated (as indicated in the Method section) or were also single-pass transmembrane proteins removed? If the latter, how many proteins were filtered out by applying this structural constraint.

4) Table 1: Could the authors flag the duplicate/multiple entries or perhaps even consolidate them

so the table contains the 258 matrisome genes and not 277 entries?

5) I think the community studying planarians may benefit from the deployment of the in-silico matrisome resource in PlanMine. Is this being envisioned?

6) My last suggestion relates to the title of the manuscript: shouldn't the term "extracellular matrix" be included in the title?

Alexandra Naba, Ph.D.

Assistant Professor – University of Illinois at Chicago

Reviewer #2:

Remarks to the Author:

The paper by Cote et al builds upon previous observations in planeria to gather evidence that muscle tissue in planerians is functionally equivalent to connective tissue in vertebrates. Previously, planeria muscle has been shown to be the source of transcripts that provide positional information to maintain regional cell identities during tissue patterning and turnover. Also previously, a collagen gene was shown to be expressed by muscle, thus suggesting that the muscle might express genes relevant for ECM formation. Finally, previous EM images suggested muscle to have secretory vesicles that may function in ECM formation.

Using transcripts and protein sequence/domain alignments, Cote et al identify a high confidence set of genes that share identify to mammalian ECM proteins. They then looked at the expression of these genes among existing single cell datasets. Although not exclusively expressed by muscle, they did perform statistical tests to show enrichment of ECM transcripts among muscle cells. They then go on to show that knockdown of a muscle-specific ECM gene disrupts the spatial locations of differentiated cells and neoblasts during homeostatic tissue turnover. This is argued to show disruption of a function that is regulated by connective tissue in vertebrates.

I found the paper very well written. The primary finding that planeria muscle cells are the primary producers of ECM components is significant for the planeria model. However, the significance of this insight for understanding connective tissue biology and evolution among metazoans is not clear. The take home message is novel but speculative and simple. The manuscript could use a bit more balance in making this connection and greater knowledge of vertebrate connective tissue and cells other than connective tissue cells that function in tissue regeneration and homeostatic turnover.

Line 12-14. May also want to point out muscles third functional role: contraction/movement? Is this an important function of muscle in planeria?

Line 45-46. At this point it would help to clarify the type(s) of muscle that harbor positional information. Is it body wall muscle only or predominantly?

Line 67. Would be nice to know here what is known about ECM structure in planeria, before focusing attention on cells that might produce ECM. It is not clear why ECM secretion is introduced within the context of vertebrate biology, and in an overly simplified way. ECM in vertebrates is ascribed predominately to the homeostatic/developmental actions of fibroblasts, to smooth muscle, and to chondrocytes and osteoblasts. There are other cellular sources of ECM in vertebrates- endothelial cells, immune cells, epidermis. These cell types are largely not present in planeria.

Line 190: Probably more accurate to write "encode protein domains"

Line 248: Need to clarify is meant by "muscle cells are the major contributor to the ECM". Is there also enrichment of proteins within muscle that function in transport/exocytosis

Line 256: How does ECM gene expression indicate that muscle secretes the majority of ECM molecules? There is an assumption in making this inference.

Line 284: What would be the criteria for defining such a fibroblast like cell in planeria? If there an objective way to do this?

Lines 289-348: Can this section be shortened as it seems a bit unbalanced relative to other sections and it is not clear that focus on a single gene and phenotype adds much to the systems-level analyses upon which the paper rests.

Line 350: The analysis of existing mouse single cell data is not very compelling as written. Are the matrisome orthologs from planeria significantly enriched in a particular mouse cell type?

Line 440-474: The discussion that attempts to link planeria muscle to vertebrate connective tissue and evolution is too speculative and simple. There are multiple types of connective tissue in vertebrates, and thus multiple types of connective tissue cells. There is likely considerable heterogeneity among cells of a presumptive connective tissue cell type. Some of the functions ascribed to vertebrate connective tissue cells could also reflect the actions of immunological cells. In the case of vertebrate regeneration, some connective tissue cells not only produce signaling molecules and ECM, they migrate and contribute to the blastema. This strikes me as a big difference between planaria muscle and vertebrate connective tissue cells.

Reviewer #3:

Remarks to the Author:

The report from Cote and colleagues defines the set of conserved domains associated with extracellular matrix biology in the planarian model system. They find many potential ECM associates genes conserved to a greater and lesser extent with genes in humans. Using single cell expression data sets they find that expression of ECM components are enriched in muscle cells, leading them to conclude that planarian muscle cells are also the "connective tissue" of planarians.

The authors then go in to characterise a very interesting function for one particular ECM component, the planarian hemicentin ortholog (hmcn-1). RNAi of this glycoprotein leads to disruption of planarian shape and structure by disrupting the ECM. An outcome of this disruption is that stem cells are able to now to leave the parenchyma and reach the margin of the animals. Using further transcriptional analysis and assessment of stem cell proliferation they demonstrate this effect is a result of ECM defects rather ant effect on stem cell biology.

I found the study to be very well written and presented for the most part, and it will be of interest to the planarian research community but also to anyone interested in the ECM and its evolution in animals. An interesting point the authors discuss is whether ECM and ECM secreting cells provide key regenerative positional information across animals, and whether this represents an evolutionarily conserved feature of animals of whether this state evolve independently.

While there are further experiments that could be added to expand the study (for example studying additional components), I see no reason for any additional experiments or major revisions as the current study makes a significant point as it is, so I would argue against unnecessary additions.

minor comments.

1. The authors might consider including reference to the recent single cells study of Zeng et al, Cell 2018
2. The authors might consider including reference to the tumour suppressor phenotype of the epigenetic regulator mll3/4 (Mihaylova et al, Nature communications, 2018) when discussing neoblasts penetrating tissue layers
3. The title of Supplementary Figure 4 is grammatically incorrect. Maybe "collagen co-expression demonstrated by FISH"?

Reviewer #1

Remarks to the Author: In this manuscript entitled "Muscle functions as a connective tissue in planarians", Cote and collaborators identify muscle, which convey positional information crucial for proper tissue regeneration, as the major ECM producing cell type in planarians. To do so, the authors first defined the "matrisome" of planarians using a bioinformatic workflow based on protein sequence analysis that was originally devised to predict the human and mouse matrisomes. They identified 258 genes encoding core matrisome or matrisome-associated proteins and further classified them in different structural/functional categories. With this tool in place, the authors next sought to identify which cell type(s) express matrisome genes in planarians. Using previously published single-cell transcriptomic data, the authors showed that the main cell type expressing matrisome genes are muscle cells. They thus propose that the muscle is the connective tissue compartment in planarians. Last, the functional relevance of muscle produced ECM components in tissue regeneration is illustrated through the example of the hemicentin1 (*hmcn1*) gene, encoding a conserved core matrisome glycoprotein. The authors demonstrate that *hmcn1* knockdown results in the mislocalization of several cell populations including neoblasts, and epidermal and nonepidermal cells, indicating that muscle produced ECM components can serve as cues to maintain proper tissular organization.

This is a very clearly written manuscript that reports not only the development of a novel resource but also proposes the new and exciting concept that the planarian muscle is a connective, ECM producing, tissue. Furthermore, it illustrates how the in-silico definition of the planarian matrisome can assist with big data annotation and the identification of novel functionally relevant genes and proteins. This publication will have without a doubt a significant impact and can appeal to a very broad readership beyond planarian aficionados, including researchers interested in studying the evolution of the ECM and ECM proteins, ECM biologists, and researchers interested in tissue regeneration.

As my expertise lies in ECM biology and per the editor's request, my few minor comments focus on the first part of the paper.

We thank the reviewer for the thorough and helpful comments on the paper and the matrisome annotation.

1) Is there a reason to suspect that other ECM proteins that do not have structurally related mammalian orthologs may be encoded by planarian genes? The example that comes to mind is the cuticular extracellular matrix of nematodes: it is an ECM that does not have a mammalian equivalent and the components of this type of ECM (cuticlins, chitins) cannot be identified using the canonical "ECM domains" originally used to define mammalian matrisomes. Can the authors comment on that?

We appreciate the reviewer pointing out that the approach of using mammalian matrisome domains may not capture all planarian ECM components. We have now addressed that planarians do not have orthologs to cuticlins or chitins in lines 111-114. We also mention that there could prove to be other planarian/spiralian/or protostome-specific ECM components that could elude detection by this *in silico* analysis in lines 351-353.

2) Page 5: Can the authors indicate how many proteins were removed by applying the size constraint of 100 amino acids. Several mammalian matrisome components, in particular belonging to the matrisome associated division and secreted factors category are smaller than 100 amino acids (a family that comes to mind is the S100 family), it would thus be interesting to briefly discuss why this size constraint was applied.

This size constraint was originally applied to reduce the number of contigs that needed manual verification of the full-length CDS, as many of these contigs represent lowly expressed transcripts, intronic sequences, assembly artifacts, or fragments of longer genes. Planarians do not contain clear members of the S100 family. Recent release in late fall 2018 of *in silico* gene predictions (Planmine 3.0) from the planarian genome greatly aided in finding full-length coding sequences. We have now inspected all 199 contigs previously removed by this size filter and identified full-length coding sequences with signal peptides. From this we found 13 additional core glycoproteins mostly with putative FN2 or lectin domains, 21 ECM regulators mainly with Kringle or other peptidase domains, and 4 small secreted factors plus 10 low confidence secreted factors with very weak domain annotations, which have now been added to Supplementary Table 1 and listed in the table below. We have clarified our annotation pipeline in the methods and listed contigs removed during the annotation process in Supplementary Table 2. All analyses in Figures 1-3, 6 and Supplementary Figures 1-3, 5 have been updated to reflect these minor changes in the underlying matrisome annotation.

gene_name	contig_id	Matrisome_category	ECM_confidence	Initial_submission
dd_12352	dd_12352	Core_glycoprotein	high	smallCDS
dd_210 (CRISPLD2)	dd_210	Core_glycoprotein	high	smallCDS
dd_2225 (LCCL)	dd_2225	Core_glycoprotein	high	smallCDS
dd_29918 (SEA)	dd_29918	Core_glycoprotein	high	noSIP
dd_33 (SFTPA1)	dd_33	Core_glycoprotein	high	smallCDS
dd_42798 (CLECT)	dd_42798	Core_glycoprotein	high	smallCDS
dd_47321 (val)	dd_47321	Core_glycoprotein	high	noSIP
dd_5456	dd_5456	Core_glycoprotein	high	smallCDS
dd_61913 (VWA)	dd_61913	Core_glycoprotein	high	smallCDS

dd_73657 (TSP1)	dd_73657	Core_glycoprotein	high	noSIP
dd_80257 (FN2)	dd_80257	Core_glycoprotein	high	smallCDS
dd_84104 (TG)	dd_84104	Core_glycoprotein	high	smallCDS
dd_96847 (CLECT)	dd_96847	Core_glycoprotein	high	smallCDS
glypican-2	dd_4078	ECM_affiliated	high	TM
glypican-3	dd_20269	ECM_affiliated	high	TM
syndecan-1	dd_4546	ECM_affiliated	high	TM
syndecan-2	dd_14098	ECM_affiliated	high	TM
dd_30891 (PLG)	dd_30891	ECM_affiliated	high	ECMregHigh
dd_101914 (DDR2)	dd_101914	ECM_regulator	high	smallCDS
dd_10224 (CTSA)	dd_10224	ECM_regulator	high	smallCDS
dd_1072 (WFIKKN2)	dd_1072	ECM_regulator	high	smallCDS
dd_12691 (WFIKKN2)	dd_12691	ECM_regulator	high	smallCDS
dd_1274	dd_1274	ECM_regulator	high	smallCDS
dd_13	dd_13	ECM_regulator	high	smallCDS
dd_17487 (TY)	dd_17487	ECM_regulator	high	smallCDS

dd_190 (WFIKKN2)	dd_190	ECM_regulator	high	smallCDS
dd_23420	dd_23420	ECM_regulator	high	TM
dd_2814	dd_2814	ECM_regulator	high	smallCDS
dd_412 (WFIKKN2)	dd_412	ECM_regulator	high	smallCDS
dd_4540 (PRSS12)	dd_4540	ECM_regulator	high	noSIP
dd_456 (CTSF)	dd_456	ECM_regulator	high	ECMregLow
dd_51 (MEP1B)	dd_51	ECM_regulator	high	smallCDS
dd_6335 (MMP28)	dd_6335	ECM_regulator	high	smallCDS
dd_73979 (CTSS)	dd_73979	ECM_regulator	high	smallCDS
dd_75096 (PRSS1)	dd_75096	ECM_regulator	high	noSIP
dd_81 (CTSCB)	dd_81	ECM_regulator	high	ECMregLow
dd_85792 (PLG)	dd_85792	ECM_regulator	high	smallCDS
dd_886	dd_886	ECM_regulator	high	smallCDS
dd_89796 (KR)	dd_89796	ECM_regulator	high	smallCDS
admp-1a	dd_38565	Secreted_factor	high	smallCDS
dd_35615	dd_35615	Secreted_factor	high	noSIP
dd_39545	dd_39545	Secreted_factor	high	noSIP
dd_97948	dd_97948	Secreted_factor	high	smallCDS
cubilin	dd_4575	Core_glycoprotein	low	noSIP
dd_1144 (colipase)	dd_1144	Core_glycoprotein	low	noSIP
dd_1789	dd_1789	Core_glycoprotein	low	noSIP
dd_194	dd_194	Core_glycoprotein	low	noSIP
dd_2336	dd_2336	Core_glycoprotein	low	noSIP
dd_636 (VWD)	dd_636	Core_glycoprotein	low	noSIP
dd_91211 (IgLDL)	dd_91211	Core_glycoprotein	low	smallCDS
dd_11820	dd_11820	Secreted_factor	low	smallCDS
dd_17258	dd_17258	Secreted_factor	low	smallCDS
dd_22273	dd_22273	Secreted_factor	low	noSIP
dd_2602	dd_2602	Secreted_factor	low	smallCDS
dd_28008	dd_28008	Secreted_factor	low	noSIP
dd_30771	dd_30771	Secreted_factor	low	noSIP
dd_43794	dd_43794	Secreted_factor	low	noSIP

dd_57981	dd_57981	Secreted_factor	low	smallCDS
dd_6028	dd_6028	Secreted_factor	low	noSIP
dd_6493	dd_6493	Secreted_factor	low	smallCDS
dd_7199	dd_7199	Secreted_factor	low	smallCDS

3) Page 5 and Method section: the authors indicate using the presence of a transmembrane domain as a method to eliminate non-matrisome proteins. However, in the mammalian matrisome, several matrisome proteins including collagens, proteoglycans (syndecans, glypicans), and proteases (MMPs) are transmembrane proteins. Were only proteins displaying 7 transmembrane domains eliminated (as indicated in the Method section) or were also single-pass transmembrane proteins removed? If the latter, how many proteins were filtered out by applying this structural constraint.

We now clarify that 196 contigs that appear to be receptors with 1 or more TM domain were removed from the analysis, except for those homologous to collagens, mmps, syndecans, glypicans with transmembrane domains (lines 474-478). These removed contigs are listed in Supplementary Table 2, which could allow future investigation of these genes. We have added information about TM domain annotation in Supplementary Tables 1 and 2.

4) Table 1: Could the authors flag the duplicate/multiple entries or perhaps even consolidate them so the table contains the 258 matrisome genes and not 277 entries?

We agree that multiple contigs per gene complicates Table 1. We have added a “representative contig” yes/no column to Table 1 and moved duplicate entries to the end of the table.

5) I think the community studying planarians may benefit from the deployment of the in silico matrisome resource in PlanMine. Is this being envisioned?

We agree that adding this as a resource to PlanMine would benefit the community and will coordinate with PlanMine to add this data post publication to their resource and other online resources. To facilitate this process we have included gene numbers (SMESG) released with PlanMine 3.0 in Supplementary Tables 1 and 2.

6) My last suggestion relates to the title of the manuscript: shouldn't the term "extracellular matrix" be included in the title?

We agree with this suggestion and have changed the title to “Muscle functions as a connective tissue and source of extracellular matrix in planarians”.

Reviewer #2

Remarks to the Author: The paper by Cote et al builds upon previous observations in planeria to gather evidence that muscle tissue in planerians is functionally equivalent to connective tissue in vertebrates. Previously, planeria muscle has been shown to be the source of transcripts that provide positional information to maintain regional cell identities during tissue patterning and turnover. Also previously, a collagen gene was shown to be expressed by muscle, thus suggesting that the muscle might express genes relevant for ECM formation. Finally, previous EM images suggested muscle to have secretory vesicles that may function in ECM formation.

Using transcripts and protein sequence/domain alignments, Cote et al identify a high confidence set of genes that share identify to mammalian ECM proteins. They then looked at the expression of these genes among existing single cell datasets. Although not exclusively expressed by muscle, they did perform statistical tests to show enrichment of ECM transcripts among muscle cells. They then go on to show that knockdown of a muscle-specific ECM gene disrupts the spatial locations of differentiated cells and neoblasts during homeostatic tissue turnover. This is argued to show disruption of a function that is regulated by connective tissue in vertebrates.

I found the paper very well written. The primary finding that planeria muscle cells are the primary producers of ECM components is significant for the planeria model. However, the significance of this insight for understanding connective tissue biology and evolution among metazoans is not clear. The take home message is novel but speculative and simple. The manuscript could use a bit more balance in making this connection and greater knowledge of vertebrate connective tissue and cells other than connective tissue cells that function in tissue regeneration and homeostatic turnover.

We thank the reviewer for a critical reading of the manuscript and suggestions for points needing clarification and further explanation.

Line 12-14. May also want to point out muscles third functional role: contraction/movement? Is this an important function of muscle in planeria?

Indeed, planarian muscle plays a key role in contractility and movement. For instance, RNAi of genes encoding contractility components, such as *tropomyosin*, result in defects that include movement and body shape defects, as well as animal lysis (Reddien et al, Dev Cell, 2005). The multitude of roles for planarian muscle are reviewed in Cebrià Front Cell Dev Biol 2016 (reference 13). We have added this background information in lines 6 and 53-54.

Line 45-46. At this point it would help to clarify the type(s) of muscle that harbor positional information. Is it body wall muscle only or predominantly?

We have clarified in lines 45-49 that whereas sub-epidermal body-wall muscle prominently expresses all position control genes (PCGs) examined in Witchley et al, Cell Reports, 2013, other muscle subtypes and other cell types also express subsets PCGs, as reported in our recent paper (Scimone et al. 2018).

Line 67. Would be nice to know here what is known about ECM structure in planeria, before focusing attention on cells that might produce ECM. It is not clear why ECM secretion is introduced within the context of vertebrate biology, and in an overly simplified way. ECM in

vertebrates is ascribed predominately to the homeostatic/developmental actions of fibroblasts, to smooth muscle, and to chondrocytes and osteoblasts. There are other cellular sources of ECM in vertebrates endothelial cells, immune cells, epidermis. These cell types are largely not present in planeria.

We now first introduce what is known about planarian ECM mainly electron microscopy observations more fully before discussing vertebrates and cell sources of ECM (lines 61-67). We have also added information about other cell sources of ECM in vertebrates, some of which were previously only in the discussion, to the introduction as suggested. The introduction of the matrisome is now placed in the context of metazoan cells that secrete ECM in lines 75-80.

Line 190. Probably more accurate to write “encode protein domains”

We have rewritten this part of the paragraph to emphasize that we are focusing on the roles and expression of genes encoding ECM core proteins on lines 190-194.

Line 248. Need to clarify is meant by “muscle cells are the major contributor to the ECM”. Is there also enrichment of proteins within muscle that function in transport/exocytosis?

We have looked at the expression of genes involved in transport and exocytosis listed in Table 3 and have looked at their broad expression in Supplementary Figures 2 and 5 (lines 247-249). Whereas muscle is not enriched in the expression of protein secretion genes involved in transport and exocytosis, many of the secretory pathway genes are expressed in muscle cells. Along with evidence from prior EM studies that muscle cells contain elaborate ER, Golgi material, and occasional ruthenium red positive secretory vesicles, it appears that cells with myofibers are competent to secrete large amounts of extracellular proteins (lines 63-67).

Line 256. How does ECM gene expression indicate that muscle secretes the majority of ECM molecules? There is an assumption in making this inference.

We have added clarification that ECM gene expression is being used as an indirect proxy for ECM secretion on lines 249-251. We have clarified that strong ECM expression is consistent with the hypothesis that muscle secretes the majority of ECM components on line 256-258.

Line 284. What would be the criteria for defining such a fibroblast like cell in planeria? If there an objective way to do this?

We think that describing the criteria more explicitly in the text is a good idea. Our approach to this was to reword the end of the Results section (line 284-6) and we have now more thoroughly discussed the argument in the discussion (lines 357-374).

In vertebrates, fibroblasts are largely described as connective tissue cells that are producers of fibrous material including collagen; these cells can divide rapidly and migrate in response to wounding (e.g., Alberts The Cell; Lodish Molecular Cell Biology). One can use these criteria to look for any cell that shares some or all of these attributes. The complete planarian cell type transcriptome atlas provides

one objective approach to looking for cell types that express ECM components like collagen. The fact that this atlas is saturated for cell types is key to enabling this approach to be systematic and as objective as possible. The major producer of collagen we identified was clearly muscle as opposed to any other cell type in the complete set planarian cell types. With regards to other attributes, the only dividing cells in planarians are neoblasts, which function as pluripotent stem cells. No differentiated cells divide. This difference between adult planarian and vertebrate biology is noted on lines 30-31, and is a clear difference between vertebrate fibroblasts and any differentiated planarian cell type. Muscle responds transcriptionally to wounding (Wurtzel et al, Dev Cell, 2015, Witchley et al, Cell Reports, 2013) and are present in the blastema (reviewed Cebrià Front Cell Dev Biol 2016). However, overall, there is no known migratory cell that moves to wounds in a behavior reminiscent of vertebrate fibroblasts. These observations together indicate the absence of a detectable differentiated planarian cell type with all hallmarks of fibroblasts. We therefore propose that aspects of connective tissue function promoted by fibroblasts in vertebrates - in particular ECM secretion - is provided instead by muscle in planarians.

Lines 289-348. Can this section be shortened as it seems a bit unbalanced relative to other sections and it is not clear that focus on a single gene and phenotype adds much to the systems level analyses upon which the paper rests.

We have shortened this section to lines 289 – 320. In addition, we have added additional phenotypes of ECM components in Supplementary Table 4 and in Supplementary Figures 6, 7 to strengthen the paper in terms of a broader investigation of ECM and muscle biology. Inhibition of ECM components and receptors likely interacting with the ECM results in a variety of phenotypes (lines 289-294, 380-383).

Line 350. The analysis of existing mouse single cell data is not very compelling as written. Are the matrisome orthologs from planaria significantly enriched in a particular mouse cell type?

We have performed the analysis of using only those vertebrate ECM genes that have orthologs in the predicted planarian matrisome and show this data in Supplementary Figure 11a. These genes are largely expressed in stromal cells (lines 332-333). We have additionally clarified that multiple vertebrate cell types express ECM components (lines 363-367) but the strongest expression is in stromal cells.

Line 440-474. The discussion that attempts to link planaria muscle to vertebrate connective tissue and evolution is too speculative and simple. There are multiple types of connective tissue in vertebrates, and thus multiple types of connective tissue cells. There is likely considerable heterogeneity among cells of a presumptive connective tissue cell type. Some of the functions ascribed to vertebrate connective tissue cells could also reflect the actions of immunological cells. In the case of vertebrate regeneration, some connective tissue cells not only produce signaling molecules and ECM, they migrate and contribute to the blastema. This strikes me as a big difference between planaria muscle and vertebrate connective tissue cells.

Our intent in the discussion is to describe the lack of a fibroblast-like cell in planarians, the fact that connective-tissue function is instead provided by muscle, and then to discuss potential evolutionary explanations for these differences and

directions that could test these evolutionary hypotheses. We have endeavored to clarify the discussion along these lines in several ways. We have added clarification that planarian neoblasts are the source of all new tissue in the blastema, which is a key difference between planarian and vertebrate regeneration strategies (lines 30-31) and references to 'dividing neoblasts' in the discussion lines 369. We have clarified that the functional similarities between vertebrate connective tissues and planarian muscle are confined to ECM production and providing positional information (lines 418-421). We have clarified that muscle responds transcriptionally to wounding and that both newly formed and pre-existing muscle cells function to orchestrate regeneration (see Scimone et al Nature 2018) on lines 394-402.

Reviewer 3

The report from Cote and colleagues defines the set of conserved domains associated with extracellular matrix biology in the planarian model system. They find many potential ECM associates genes conserved to a greater and lesser extent with genes in humans. Using single cell expression data sets they find that expression of ECM components are enriched in muscle cells, leading them to conclude that planarian muscle cells are also the "connective tissue" of planarians.

The authors then go in to characterise a very interesting function for one particular ECM component, the planarian hemicentin ortholog (*hmcn1*). RNAi of this glycoprotein leads to disruption of planarian shape and structure by disrupting the ECM. An outcome of this disruption is that stem cells are able to now to leave the parenchyma and reach the margin of the animals. Using further transcriptional analysis and assessment of stem cell proliferation they demonstrate this effect is a result of ECM defects rather ant effect on stem cell biology.

I found the study to be very well written and presented for the most part, and it will be of interest to the planarian research community but also to anyone interested in the ECM and its evolution in animals. An interesting point the authors discuss is whether ECM and ECM secreting cells provide key regenerative positional information across animals, and whether this represents an evolutionarily conserved feature of animals or whether this state evolve independently.

While there are further experiments that could be added to expand the study (for example studying additional components), I see no reason for any additional experiments or major revisions as the current study makes a significant point as it is, so I would argue against unnecessary additions.

We thank the reviewer for the careful reading of the manuscript and supportive remarks. We went ahead and added more information about additional components as suggested. This information is in Supplementary Table 4, Supplementary Figures 6, 7 and described in the text on lines 289-294, 380-383.

minor comments

1. The authors might consider including reference to the recent single cells study of Zeng et al, Cell 2018.

We have added citations to Zeng et al, Cell 2018 (ref. 6) to lines 197-198 as this study shows significant uses of planarian single-cell data to characterize cell types.

2. The authors might consider including reference to the tumour suppressor phenotype of the epigenetic regulator *mlI3/4* (Mihaylova et al, Nature communications, 2018) when discussing neoblasts penetrating tissue layers.

We thank the reviewer for pointing out this study and its relevance to neoblast biology. We have added the citation to line 305.

3. The title of Supplementary Figure 4 is grammatically incorrect. Maybe "collagen coexpression demonstrated by FISH"?

We thank the reviewer for catching this mistake and have changed the title of Supplementary Figure 4 to “Planarian collagen co-expression demonstrated by FISH”.

Reviewers' Comments:

Reviewer #2:

Remarks to the Author:

I appreciate the responsiveness of the authors to reviewer concerns. The manuscript is much improved and I have no further comments/concerns.